# MicroRNA Profiling of the Inflammatory Response after Early and Late Asthmatic Reaction

**DOI:** 10.3390/ijms25021356

**Published:** 2024-01-22

**Authors:** Ruth P. Duecker, Oguzhan Alemdar, Andreas Wimmers, Lucia Gronau, Andreas G. Chiocchetti, Eva M. Valesky, Helena Donath, Jordis Trischler, Katharina Blumchen, Stefan Zielen, Ralf Schubert

**Affiliations:** 1Department of Pediatrics, Division of Pneumology, Allergology, Infectious Diseases and Gastroenterology, University Hospital, Goethe University Frankfurt, 60590 Frankfurt am Main, Germany; oguzhan.alemdar@web.de (O.A.); a.wimmers@medaimun.de (A.W.); gronau@med.uni-frankfurt.de (L.G.); donath@med.uni-frankfurt.de (H.D.); jordis.trischler@gmail.com (J.T.); bluemchen@med.uni-frankfurt.de (K.B.); s.zielen@medaimun.de (S.Z.); r.schubert@med.uni-frankfurt.de (R.S.); 2Respiratory Research Institute, Medaimun GmbH, 60596 Frankfurt am Main, Germany; 3Department of Child and Adolescent Psychiatry, Psychosomatics and Psychotherapy, University Hospital, Goethe University Frankfurt, 60590 Frankfurt am Main, Germany; andreas.chiocchetti@med.uni-frankfurt.de; 4Department of Dermatology, Venerology and Allergology, University Hospital, Goethe University Frankfurt, 60590 Frankfurt am Main, Germany; eva.valesky@t-online.de

**Keywords:** epigenetic regulation, microRNAs, early asthmatic and late asthmatic reaction, bronchial allergen provocation, house dust allergy

## Abstract

A high proportion of house dust mite (HDM)-allergic asthmatics suffer from both an early asthmatic reaction (EAR) and a late asthmatic reaction (LAR) which follows it. In these patients, allergic inflammation is more relevant. MiRNAs have been shown to play an important role in the regulation of asthma’s pathology. The aim of this study was to analyze the miRNA profile in patients with mild asthma and an HDM allergy after bronchial allergen provocation (BAP). Seventeen patients with EAR/no LAR and 17 patients with EAR plus LAR, determined by a significant fall in FEV1 after BAP, were differentially analyzed. As expected, patients with EAR plus LAR showed a more pronounced allergic inflammation and FEV1 delta drop after 24 h. NGS-miRNA analysis identified the down-regulation of miR-15a-5p, miR-15b-5p, and miR-374a-5p after BAP with the highest significance in patients with EAR plus LAR, which were negatively correlated with eNO and the maximum decrease in FEV1. These miRNAs have shared targets like *CCND1*, *VEGFA*, and *GSK3B*, which are known to be involved in airway remodeling, basement membrane thickening, and Extracellular Matrix deposition. NGS-profiling identified miRNAs involved in the inflammatory response after BAP with HDM extract, which might be useful to predict a LAR.

## 1. Introduction

Allergic asthma is associated with seasonal or perennial allergen exposure [1,2]. Bronchial allergen exposure in allergic asthmatics leads to many physiological and inflammatory manifestations of asthma, including airway obstruction, eosinophilic inflammation, and an increase in bronchial hyperresponsiveness (BHR) [3,4,5,6]. Several clinical associations underline the hypothesis that inhaled allergen exposure leads to clinical exacerbations. There is a high asthma morbidity and mortality associated with the inhalation of high *Alternaria* concentrations in the USA [7,8], increased pollen exposure during thunderstorms [9], and the inhalation of soybean dust in Spain [10].

Bronchial allergen inhalation provokes an early asthmatic reaction (EAR) in sensitized asthmatics within minutes, which subsides within 1–3 h [5,6]. About 50–80 percent of asthmatics experience a longer-lasting late asthmatic reaction (LAR) that peaks within 3–9 h and resolves within 24 h [11,12]. The cellular mechanisms of EAR and LAR have been extensively studied: EAR is mediated by the rapid release of preformed and newly formed lipid mediators such as leukotrienes, prostaglandins, and the cytokines IL-4, IL-5, and IL-13 via the cross-linking of membrane-bound IgE of local mast cells [12,13]. Bronchoconstriction, hypersecretion, and swelling of the bronchial mucosa could be also detected [12,14].

In contrast to EAR, LARs lead to mucosal inflammation by T lymphocytes, Treg cells, and eosinophils [12,14,15]. This leads to the further recruitment inflammatory cells, especially Th_2_ cells and eosinophilic granulocytes, but also basophilic and neutrophilic granulocytes and macrophages [16,17]. Systemic and bronchial eosinophilic inflammation can be detected 24 h after bronchial allergen exposure [5,6]. The induced eosinophilic inflammation correlates with an increase in exhaled NO (eNO). Likewise, the BHR, which is closely related to the inflammatory response of the LAR, increases for several days [6,18,19,20]. This shows that, in patients with allergic asthma, allergen exposure not only causes acute bronchospasms, but also triggers a longer-lasting cellular inflammatory response that can lead to increased BHR and asthma severity [12,21]. Recurrent or persistent allergen exposure leads to the development of chronic inflammation and structural remodeling processes [12,22]. 

However, it is not well understood why some allergic patients develop an EAR only, whereas others respond with a dual asthmatic reaction—EAR plus LAR upon, for example, bronchial allergen provocation (BAP). To better understand the cellular and molecular mechanisms and why not all individuals with allergic asthma develop a LAR, RNA sequencing of total RNA was performed by Singh et al. [23] to detect biomarkers that are predictive of a LAR. The authors identified a biomarker panel, the Trinity biomarker panel, consisting of known and novel biomarker transcripts. The Trinity biomarker panel proved useful for predicting the response of individuals exhibiting a dual response (EAR plus LAR) with an accuracy between 0.70 and 0.75. Taken together, these results indicate a dysregulation of the complex immune signaling network that leads to a LAR in patients with allergic asthma. Possibly, non-coding pieces of RNA that bind to messenger RNAs (messenger RNAs, mRNAs), so-called microRNAs (miRNAs), can provide additional insights into the biological mechanisms of a LAR.

Several studies on chronic or allergic asthma indicate that epigenetic regulation contributes to the inflammatory response observed after allergen exposure [24,25,26]. Post-transcriptional regulation by miRNAs is an important central epigenetic mechanism. miRNAs are short, highly conserved, non-coding ribonucleic acids that play an important role in the complex network of gene regulation, particularly gene silencing. Thus, miRNAs play an important role in the control of a variety of cellular processes such as cell proliferation, cell differentiation, and apoptosis [27,28]. miRNAs are known to modulate the acute phase of inflammation, making them central key players in understanding the induction of EAR and LAR after allergen exposure [26,29]. 

Recent studies have highlighted the potential of circulating miRNAs as non-invasive biomarkers for allergic asthma. For instance, in pediatric populations, 122 circulating miRNAs were identified to differentiate between asthmatic and non-asthmatic children, showcasing their discriminatory power. Studies by The Childhood Asthma Management Program (CAMP) further unveiled specific miRNAs associated with lung-function parameters and bronchial hyper-responsiveness, paving the way for their utilization in predicting asthma outcomes [30,31]. In adult asthma, distinct miRNA profiles have been observed. Exhaled breath condensates from asthmatic individuals revealed the significantly decreased expression of let-7a, miR-21, -133a, -155, -328, and -1248 compared to healthy subjects. These miRNAs were found to target numerous type 2 mediators, indicating their involvement in asthma’s pathogenesis [32]. Machine learning approaches applied to miRNA expression data from CAMP cohort studies suggested a combination of miRNAs, including miR-146b, miR-206, and miR-720, as potential predictive markers for asthma [31].

Despite this growing body of knowledge, a comprehensive investigation into the miRNA profiling specifically associated with EARs and LARs in patients with a house dust mite (HDM) allergy after BAP has been lacking. Our study seeks to bridge this gap by exploring the distinct miRNA signatures linked to EARs and LARs in patients a with HDM allergy after BAP in the context of allergic inflammation.

## 2. Results

There was no difference between patients exhibiting only an EAR and those experiencing both an EAR plus LAR concerning age, sex, forced vital capacity (FVC), forced expiratory volume in one second (FEV1), or the outcomes of the methacholine challenge. However, patients with EAR plus LAR demonstrated significantly higher levels of HDM-specific IgE compared to those with only EAR/no LAR; *p* < 0.05. The clinical patient characteristics are summarized in Table 1.

### 2.1. Bronchial Allergen Provocation

All 34 patients exhibited a significant EAR characterized by a maximum fall in FEV1 of >20% after BAP. Additionally, a significant LAR was observed in 17 patients, with a maximum fall in FEV1 of >15%, as illustrated in Figure 1a. Interestingly, there was no significant difference caused by the eliciting dose of HDM extract used for BAP between the groups (HDM extract dose: EAR/no LAR: median 57.6 SBE/mL (range: 6.4–248.1) vs. EAR plus LAR median 87 SBE/mL (range 5.7–630; n.s.).

The maximum fall in FEV1 during EAR did not exhibit significant differences between the groups (EAR/no LAR: 31.85 ± 6.35% vs. EAR plus LAR: 36.57 ± 13.5%; n.s.; Figure 1b). However, the maximum fall in FEV1 within 6–9 h after BAP was notably higher in the EAR plus LAR group (EAR/no EAR: 6.36% ± 3.4% vs. EAR plus LAR: 31.7% ± 10.29%; *p* < 0.0001; Figure 1c). 

As anticipated, a robust inflammatory response was evident 24 h after BAP, marked by a significant increase in eNO and peripheral blood eosinophilia in both groups (Appendix A). While some results exhibited overlap, the delta change, comparing the response 24 h after BAP with the baseline, consistently showed a more pronounced effect in the EAR plus LAR group for both eNO (EAR/no LAR: median 23.88 ± 8.85 ppb vs. EAR plus LAR: 46.53 ± 7.50 ppb; *p* < 0.01; Figure 2a) and delta FEV1 decrease (EAR/no LAR: 6.47 ± 51.08 mL vs. EAR plus LAR: −357.60 ± 75.85 mL; *p* < 0.01; Figure 2c). In contrast, there were no differences in the delta eosinophil counts in peripheral blood between the groups (Figure 2b).

### 2.2. miRNAs

#### 2.2.1. Influence of BAP on miRNA Expression

To assess microRNA (miRNA) profiles, the global miRNA expression was analyzed in peripheral blood 7 h after BAP, comparing patients with EAR/no LAR to those with both EAR and LAR (EAR plus LAR). The analysis included eight patients in each group and eight healthy controls, resulting in approximately 157 million reads with an average of 2,371,777 reads per sample. A total of 626 miRNAs were detected. The groups exhibited no differences (chi-square test *p* values > 0.1) in age, sex, RNA quality, or read depth. Statistical analysis identified 62 dysregulated miRNAs (and 10 piRNAs) with an uncorrected *p* < 0.05 in the EAR/no LAR group, of which 6 miRNAs were dysregulated with an uncorrected *p* < 0.001, 1 was downregulated and 5 were upregulated (Figure 3a, Appendix A). In the EAR plus LAR group, 99 miRNAs (and 5 piRNAs) were dysregulated with a *p* < 0.05, including 12 miRNAs with a *p* < 0.001 compared to controls (Figure 3b, Appendix A), 6 of which were downregulated and six of which were upregulated. Comparing all significantly dysregulated miRNAs (*p* < 0.05) in both groups, a Venn diagram analysis revealed 36 miRNAs that were significantly dysregulated in the EAR/no LAR group only, 4 with a significance of *p* < 0.001, namely miR-708-5p, mir-4746-5p, miR-5100, and mir-6849-3p (Figure 3c). In the EAR plus LAR group, 68 miRNAs were significantly dysregulated, with 10 reaching significance of *p* < 0.001, including miR-15a-5p, miR-15b-5p, miR-32-5p, miR-96-5p, miR-146a-5p, miR-374a-5p, miR-378c, miR-1299, miR-4452, and miR-6832-3p. Among the 36 miRNAs that were dysregulated in both groups, only miR-4436 reached the significance *p* < 0.001. Of the 10 miRNAs that were significantly dysregulated only in the EAR plus LAR group, 3 were successfully confirmed by TaqMan qPCR: miR-15a-5p, miR-15b-5p, and miR-374-5p (Figure 3d–f). Interestingly, the expression levels of all three miRNAs tended to be downregulated in the EAR/no LAR group as well.

#### 2.2.2. KEGG-Pathway and Target Analysis

To elucidate the impact of miRNAs on biological processes, we conducted KEGG-pathway enrichment analysis and evaluated the targets of specific miRNAs. Collectively, miR-15a-5p, miR-15b-5p, and miR-374a-5p were predicted to be involved in 24 pathways, regulating a total of 1120 targets. Among these pathways, several are pertinent to asthma and inflammation, including TGF-β signaling (*p* < 0.00113), focal adhesion (padj. < 0.00325), Toll-like receptor signaling (*p* < 0.0117), and T cell receptor (*p* < 0.0128) signaling. Notably, 35 targets were identified as interacting with all three miRNAs (Figure 4a, Appendix A).

The analysis of miRNA targets within specific pathways revealed the regulation of 15 targets in TGF-β signaling, 14 targets in Toll-like receptor signaling, 26 targets in focal adhesion, and 14 targets in T cell receptor signaling (Figure 4b–e). Notably, three common target genes—cyclin D1 (CCND1), vascular endothelial growth factor A (VEGFA), and glycogen synthase kinase-3 beta (GSK3B)—were identified as being regulated by all three miRNAs within pathways associated with asthma and inflammation.

#### 2.2.3. Correlation Analyses

Correlation analyses were conducted to explore the relationship between circulating miRNA expression in the blood and inflammatory parameters such as eNO and eosinophils measured in peripheral blood, as well as a lung function parameter—the decrease in FEV1 after BAP. The expression of miRNAs was significantly correlated with eNO (miR-15a-5p: *r* = −0.360; *p* < 0.05, miR15b5p: *r* = −0.385; *p* < 0.05, miR347a5p: *r* = −0.332; *p*  <  0.05) and the maximum fall in FEV1 7–9 h after BAP (miR-15a-5p: *r* = −0.520; *p* < 0.001, miR-15b-5p: *r* = −0.468; *p* = 0.01, miR347a5p: *r* = −0.510; *p* < 0.001; Figure 5a,c). In contrast, no correlation was detected between the miRNAs and eosinophil counts in peripheral blood (Figure 5b).

## 3. Discussion

BAP stands as a well-established diagnostic cornerstone and is widely acknowledged as the gold standard for investigating allergic asthmatic reactions [5,33,34]. The focal point of our study was to discern potential distinctions in the inflammatory response and miRNA regulation between patients manifesting an early asthmatic reaction only (EAR/no LAR) and those experiencing a dual asthmatic response, characterized by an early reaction followed by a late asthmatic reaction (EAR plus LAR). As anticipated, individuals with an EAR plus LAR exhibited a notably heightened blood-eosinophilic and eNO inflammatory response compared to those with an EAR/no LAR. Additionally, the decline in the FEV1 between 3 and 9 h after BAP was significantly more pronounced in the EAR plus LAR group. 

This observation aligns with existing knowledge. The detectable systemic and bronchial eosinophilic inflammation 24 h post-BAP, correlating with exhaled NO, is well documented [6,17,18,19,35]. Furthermore, symptoms associated with LAR tend to be more severe, encompassing heightened mucus production, inflammation (swelling), and structural changes in the airways, concomitant with an augmented bronchial hyperresponsiveness (BHR)—a pivotal characteristic of asthma [12,36,37]. Consequently, the LAR is deemed an integral facet of the asthma process, even in cases categorized as mild disease [38,39].

In a study conducted by Lopuhaä et al., 52 HDM-allergic patients were investigated, with 26 individuals experiencing asthma and the remaining 26 with perennial rhinitis. All patients underwent testing using HDM [38]. The study yielded insightful conclusions: The EARs exhibited similarities between patients with asthma and non-asthmatics. In contrast, the LAR was more severe in asthmatics, as evidenced by a greater decrease in FEV1 (asthma—median 27.6% vs. non-asthmatics—median 18.9%, *p* = 0.02). Furthermore, BAP induced significant increases in sputum eosinophils and eosinophil cationic protein (ECP), albeit with a considerable overlap in both groups. It is important to note that the study’s selection criteria relied on clinical signs alone, characterizing patients post hoc with BAP. In contrast, our study compared selectively chosen patients with only EAR to those with both EAR and subsequent LAR. 

The clinical significance of the LAR becomes particularly evident when evaluating the inflammatory response following BAP with pharmacological intervention [5]. Inhaled steroids (ICS) stand as the primary anti-inflammatory drugs for controlling asthmatic inflammation. Interestingly, the EAR is minimally or not affected by ICS [40]. Moreover, anti-IL-5 monoclonal antibodies such as mepolizumab and reslizumab exhibit a similar anti-inflammatory profile. These antibodies demonstrate no impact on the EAR but significantly attenuate the LAR [41]. The exceptional specificity of the LAR as a distinctive feature of allergic asthma, underscoring its greater importance in comprehending and intervening in the immunologic diathesis, was recently emphasized in the early clinical development program of a more potent anti-IgE antibody, ligelizumab [41]. Despite ligelizumab demonstrating a superior efficacy over omalizumab in inhibiting the high-affinity FcεRI receptor and mitigating the EAR, it did not yield the desired effectiveness in a phase 2 study involving severe uncontrolled asthma [42]. This outcome is most likely attributed to ligelizumab’s inefficiency in blocking the function of the FcεRII/CD23 receptor, which is responsible for the inflammatory response in asthma, as indicated by in vitro data coupled with drug concentration analysis [41]. Therefore, it is imperative for a new asthma drug or monoclonal antibody to exhibit efficacy in blocking LAR or demonstrating anti-inflammatory potency to be effective in severe allergic asthma. Additionally, these studies indicate that the LAR is a distinct entity and does not necessarily subside when a significant EAR is detectable.

Consistent with these observations, NGS analysis revealed 68 dysregulated miRNAs in the EAR plus LAR group. Among these, miR-15a-5p, miR-15b-5p, and miR-374a-5p exhibited the most significant downregulation and demonstrated a negative correlation with eNO and the maximum decrease in FEV1 after BAP. 

Numerous miRNAs, including miR-146a, -21, 126, -139, -15b, -186, -342, -374a, -409, -660, -942, and, -1290, -142, and 191, have been proposed as crucial regulators in asthma, showing correlations with lung function parameters [31,41,43,44]. A substantial proportion of these miRNAs, such as miR-146a, -21, 126, -139, -15b, -374a, and -142, were found to be significantly dysregulated in the current patient cohort. Of particular scientific interest is the observed downregulation of miR-15b and miR-374a following BAP in conjunction with LAR. For instance, previous reports have indicated reduced expressions of miR-15b-5p in the plasma of asthmatics. Furthermore, miR-15b-5p has been implicated in regulating the proliferation, migration, inflammatory response, and extracellular matrix (ECM) deposition of airway smooth muscle (ASM) cells [41,45,46]. Research has demonstrated that miR-15a-5p plays a role in downregulating the production of inflammatory chemokines, such as IL-10, and moderates T lymphocyte function [47,48,49]. Similarly, an anti-inflammatory effect has been attributed to miR-374a-5p. The expression of miR-374a-5p was found to be reduced in the serum of patients with sepsis, with a positive association observed between miR-374a-5p expression and the downregulation of pro-inflammatory cytokines [50,51]. In the context of allergic asthma, miR-374a exhibited a significant decrease in expression in individuals who were naïve to ICS and showed a correlation with lung function [44,52].

We identified an additional miRNA, miR15a, which was significantly downregulated in the EAR plus LAR group. Previous studies have indicated that lower concentrations of miR-15a lead to the decreased expression of vascular endothelial growth factor A (VEGFA), an entity found to be over-expressed in the sputum and serum of patients with asthma [53]. In line with this, Nakano et al. reported that miR-15a regulates VEGFA expression in peripheral blood CD4^+^ T cells and is significantly decreased in pediatric patients with asthma [54]. Furthermore, a recent protein–protein interaction network analysis identified miR-15a as an important target that may hold therapeutic potential for children’s allergic asthma therapy [55]. 

miRNAs have been shown to be promising biomarkers for asthma diagnosis and disease monitoring in a variety of studies [52,56]. In respect to our findings, Milger et al. described a combination of five miRNA ratios (miR-21-5p/miR-15a-5p; miR-27a-3p/miR-15a-5p; miR-29c-3p/miR15a-5p; miR-223-3p/miR-425–5p; miR-15a-5p/miR342–3p) as a reliable biomarker in allergic asthma [57]. Interestingly, of these miRNAs, we found miR-15a-5p, miR-21-5p, miR-27a-3p, miR-29c-3p, and miR-126 dysregulated in the EAR plus LAR group. In line with Coskunpinar et al., we also found miR-142-5p, miR-376c-3p, miR-27b-3p, miR-26b-5p, miR-15b-5p, and miR-29c-3p deregulated in the pathogenesis of asthma [58]. Additionally, the altered expression of miR-374a was observed in allergic asthmatics in response to inhaled corticosteroid usage by Weidner et al. [52]. Thus, miR-15a-5p, miR-15b-5p, and miR-374a are already described in relation to asthma and might have diagnostic value in determining affection status, severity, and treatment response.

The dysregulation of the miRNAs identified in this study appears to play a role in the regulation of inflammatory processes in the lung. Target analysis of the miRNAs in the individual pathways revealed three shared targets: Cyclin D1 (CCND1), VEGFA, and glykogensynthase-Kinase 3(GSK3b). All three targets are implicated in airway remodeling, leading to structural changes involving airway smooth muscle (ASM) proliferation and hypertrophy, basement membrane thickening, and ECM deposition [59,60]. Cyclin D1 (CCND1) is a crucial regulator for cell proliferation and has been found to be upregulated in the airway smooth muscle of individuals with asthma, as well as being associated with airway remodeling in animal models [61]. Additionally, Li et al. reported a significant association of CCND1 promoter genotypes with asthma susceptibility in the Taiwanese population [62]. As mentioned earlier, the activation of VEGFA contributes to epithelial barrier dysfunction, airway inflammation, and the release of T-helper 2 cytokines through the activation of the VEGFA/VEGFR2 signaling pathway [53,54]. 

Glycogen synthase kinase-3 (GSK-3) is a protein kinase that plays a pivotal role in regulating various critical cellular functions, and increased GSK-3 activity has been implicated in inflammatory diseases, including asthma [63]. Bentley et al. observed a correlation between airway smooth muscle hyperplasia and hypertrophy correlated with GSK-3 (beta) phosphorylation in a mouse model of asthma [64]. In a study by Bao et al., GSK-3 inhibition was found to attenuate allergic inflammation in mice [65]. Additionally, the inhibition of GSK-3b activity has been demonstrated to induce a hypertrophic phenotype in human ASM cells, as evidenced by morphological, biochemical, and functional criteria [66].

Given the influence of miR-15a/b and miR-374a on the regulation of airway smooth muscle cell proliferation and inflammation, the addition of these miRNAs, as miRNA mimetics, could influence the progression of airway remodeling by downregulating the expression of their targets. In this regard, a therapeutic effect of oligo-single-stranded DNA mimicking miR-15a-5p on multiple myeloma cells through downregulating the expression level of VEGF-A has been already shown [67]. Moreover, microRNA-374a suppresses the progression of colon cancer by directly reducing CCND1 to inactivate the PI3K/AKT pathway [68]. Furthermore, Sun et al. demonstrated that the miR-26b mimic inhibited inflammation cell proliferation associated with rheumatoid arthritis by targeting Cyclin D and down-regulating GSK-3β expression [69]. Thus, applying this knowledge to asthma research, investigating the specific miRNAs involved in the pathology of asthma could uncover new avenues for therapeutic interventions.

Nevertheless, it is important to acknowledge the limitations of the present study. Due to the study design, test subjects with an HDM below <0.75 sIgE KU/E or no bronchial hyperreactivity < 2mg methacholine were included as controls. This study primarily aimed to identify differences in the miRNA response between EAR plus LAR versus EAR/without LAR patients. Thus, the reference group was used for baseline comparison. To exclude potential biases driven by the reference group, miRNAs differentially up- and down-regulated in the same direction in both the EAR plus LAR and the EAR/without LAR vs the reference were excluded. As is evident from the results, there was significant overlap in eNO and eosinophil increases after 24 h in patients exhibiting solely EAR/no LAR compared to those with a dual response of EAR plus LAR. This overlap may be attributed to our patient selection criteria. As illustrated in Figure 1, nine patients with an HDM allergy lacking BHR were excluded from the study. These patients exclusively had allergic rhinitis and, if included, might have been more likely to exhibit no signs of LAR. Additionally, this study’s sample size was relatively small, and local inflammatory responses were not assessed through sputum analysis. It is noteworthy that few studies have focused on the impact of steroid therapies on miRNA regulation in allergic diseases [70]. In a modest cohort of 16 adult asthmatics, ICS therapy demonstrated a modest effect on pathological miRNA expression patterns in bronchial epithelial brushings alternating nine miRNAs [71].

Elevated levels of circulating miR-21 were identified in a pediatric cohort of ICS-resistant asthmatics compared with ICS responders [72]. However, the miR-21 levels were comparable between ICS-resistant children and asthmatic children not on ICS treatment. This suggests that the modified miRNA profile may be contingent on the level of asthma control. Similarly, miRNA expression showed correlation with use of systemic corticosteroids and leukotriene receptor antagonists (LTRA) [57].

In summary, miRNA profiling revealed distinct miRNA signatures involved in the inflammatory response following bronchial allergen provocation (BAP) with HDM extract, comparing patients with EAR only to those with both EAR and LAR. Among these miRNAs, miR-15a/b-5p and miR374a-5p exhibited the highest significance and displayed a negative correlation with eNO and the maximum decrease in FEV1 after BAP. Importantly, these three miRNAs shared common targets including CCND1, VEGFA, and GSK3B, known for their involvement in airway remodeling. Patients experiencing EAR plus LAR are more susceptible to developing chronic persistent asthma. However, further studies are warranted to explore the potential utility of miRNAs in predicting LARs and to investigate their potential role in the development of future miRNA-based therapeutics in this context. 

## 4. Materials and Methods

### 4.1. Study Design

This study was conducted as part of the AMG study titled “Effects of tregalizumab on allergen-induced airway responses and airway inflammation in asthmatic patients” (Eudrac-CT number 2020-000585-41). This study involved the precise characterization of 89 patients with mild episodic asthma and HDM-allergy. Approval for this study was obtained from the Ethics Committee of the State Medical Association of Hessen. Additionally, this study was registered with the European trials register (https://www.clinicaltrialsregister.eu; accessed on December 2020). After receiving comprehensive information about the study and providing written consent, all patients underwent various assessments during their first visit, including a skin prick test (SPT) with HDM extract, a lung function test, a methacholine challenge, an assessment of eosinophil count from peripheral blood, and measurement of HDM-specific IgE. Patients with confirmed BHR of less than 2 mg methacholine and an HDM-specific IgE RAST class greater than 2 were invited to visit 2. During the second visit, participants underwent lung function measurement, eNO determination, a titrated BAP with HDM extract, and microRNA measurements 7 h after BAP. At the third visit, conducted 24 h after BAP, another lung function test was conducted, and measurements for eNO and eosinophils count from peripheral blood were taken. The study flow chart, illustrating these stages, is provided in Figure 6. For the current study, 17 patients with EAR/no LAR were compared with 17 matched patients with EAR plus LAR. Participants had to meet specific inclusion and exclusion criteria:

Inclusion Criteria
Written informed consent from study participants;Age between 18 and 65 years;Known sensitization to HDM (RAST class > 2);Bronchial hyperreactivity (BHR) measured by methacholine challenge PD 20 FEV1 < 2 mg methacholine;BMI < 30.

Exclusion criteria
Age < 18 years;Lung function FVC < 80% and FEV1 < 75%;Other chronic diseases or infections (e.g., HIV, TB);Pregnancy;Treatment with inhaled and systemic corticosteroids;Known alcohol, drug, and/or medication abuse;Inability to understand the scope of the study.


Controls were defined as test subjects which did not meet the inclusion criteria. They were seen for visit 1 like all other subjects; however, they did not qualify for visit 2 (with BAP) due to several reasons: Five out of eight controls had a specific IgE to HDM ratio below <0.75 sIgE KU/E (below Rast class 2) and the remaining three had no bronchial hyperreactivity < 2mg methacholine.

Sample size estimation was based on two previous studies by our group [42,73]. Considering a standard deviation of 11% for the maximum FEV1 decline in both groups (group 1 with EAR only and group 2 with EAR plus LAR), α = 0.05, the two-sided *t*-test requires 14 subjects in each arm to obtain a power of 90% to detect a difference in baseline-adjusted FEV1 of 12.5 percentage points [73]. In addition, the assumption was that the EAR and LAR standard deviation was similar, approximately 11% after BAP, as evidenced in the publication by Trischler et al. [42]. In this study, we could demonstrate a significant modification of the EAR and LAR by intervention with omalizumab in 10 patients only. Thus, based on both previous studies, we estimated that n = 14 patients per group for both *t*-tests would achieve a power of approx. 90% to detect a 14% difference in LAR between group 1 with EAR only and group 2 with EAR plus LAR. To allow for dropouts, n = 17 patients were included in each group.

### 4.2. Skin Prick Test

A skin prick test (SPT) was conducted using standard allergens, including grass and rye pollen, alder hasel, birch pollen, *Dermatophagoides farinae* (Dfa) and *Dermatophagoides pteronyssinus* (Dpa), and cat dander (Allergopharma, Rheinbeck, Germany). The mean of the largest diameter of the wheal was measured for each allergen. A response of at least 3 mm was considered positive, indicating sensitivity or reactivity to the specific allergen.

### 4.3. Measurement of Exhaled NO

For measurements of eNO, the NIOX Vero^®^ (NIOX Nitric Oxide Monitoring System, Circassia AB, Bad Homburg, Germany) was employed. The measurement procedure followed the guidelines set by the American Thoracic Society (ATS) [74].

### 4.4. Pulmonary Function Testing

Baseline pulmonary function tests were conducted using the MasterScreen spirometer (Vyaire Medical GmbH, Höchberg, Germany). The recorded parameters included: Forced vital capacity (FVC), forced expiratory volume in 1 s (FEV1), and the FEV1/FVC. The methacholine challenge was performed using an aerosol provocation system (APS, MedicAid-dosimeter; CareFusion, Germany) as previously described [6].

### 4.5. Bronchial Allergen Provocation

The aerosol provocation system (APS) dosimeter technique (Vyaire Medical GmbH, Höchberg, Germany) was employed for the BAP, following recent descriptions [33,73]. Before the BAP, lyophilized HDM allergen (*D. far.*) (approval number 467/a/87b, batch: T7005057-DE (Allergopharma KG, Reinbek, Germany)) was dissolved in 5 mL of 0.9% saline. The challenge solution contained a concentration of 5000 standardized biological units (SBE)/mL. The allergen solution was nebulized in six incremental steps, doubling the dose at each step, with a starting dose of 10 allergen units (SBE). Spirometry was performed 10 min after each dose until a significant decrease in the FEV1 was recorded. An EAR was defined as a decrease of 20% in FEV1. The cumulative dose was determined by adding up the inhaled allergen doses. FEV1 was measured by spirometry for up to 10 hours following EAR, and a LAR was defined as a decrease in FEV1 ≥ 15%, occurring 6–9 h after BAP. 

### 4.6. Laboratoy Measurements

Serum was analyzed for total *Dermatophagoides pteronyssinus* (Dpt)—specific I and *Dermatophagoides farinae* (Dfa)—specific IgE by the ImmunoCAP system (Thermo Fisher Scientific, Dreieich, Germany).

### 4.7. miRNA and Next-Generation Sequencing

Total RNA, including miRNA, was isolated from peripheral blood samples using the Paxgene Blood miRNA Kit (Qiagen, Hilden, Germany) according to the manufacturer’s instructions. RNA concentration was determined using Nanodrop Lite spectroscopy (Thermo Scientific, Dreieich, Germany), and RNA integrity (RIN) was assessed using the Agilent RNA 6000 Nano Kit and Agilent 2100 Bioanalyzer (Agilent Technologies, Santa Carla, CA, USA). miRNA libraries were prepared using the QIAseq miRNA Library Kit (Qiagen, Hilden, Germany).

cDNA concentrations were determined using the Qubit dsDNA Assay Kit, Qubit Assay Tubes, and a Qubit 3.0 Fluorometer (all from Thermo Fisher Scientific, Dreieich, Germany). DNA quality was checked using Agilent High Sensitivity DNA Reagents with High Sensitivity DNA Chips and an Agilent 2100 Bioanalyzer (Agilent Technologies, Santa Carla, CA, USA). Libraries were frozen at −20 °C until miRNA sequencing.

Next-generation sequencing (NGS) was performed using MiSeq Reagent Kit v3, PhiX Sequencing Control v3, and MiSeq™ Desktop Sequencer (Illumina Inc., San Diego, CA, USA). Data were preprocessed using the Qiagen Web Portal Service, which converts coverage files into raw count matrices (accessed on 23 August 2023). Differential expression analysis was performed in RStudio 1.2.1335 (https://cran.r-project.org/, accessed on 4 September 2023) as previously described [75]. 

Coverage files were converted into raw count matrices using the Qiagen pipeline. Data were analyzed for differences in age, sex, RNA quality, and read depth by chisq. Test. 

No a priori filtering for sparse read counts or normalization has been applied, since the implemented pipeline accounts for low reads internally. To identify technical outliers, we performed hierarchical cluster analysis of raw count data based on the Euclidean distance between samples and the Ward algorithm implemented in R using the normalized read counts (function counts (Reads, normalize = T)) of miRNAs or the top 100 miRNAs based on variance.

For comparisons between two groups, we loaded the respective raw read counts into DESeq2 using the “DESeqDataSetFromMatrix” function. Differential expressions were estimated using the function “DESeq” with fit-typ = “local” based on the developer’s recommendation.

Since groups showed no differences (ANOVA *p*-values > 0.1) with respect to confounders (RIN, ConcBioAn, Rati0260280, Rati0260230, ConcQubit, Barcode) and read-depth, no additional corrections were to be included.

Fdr correction (Benjamini Hochberg) was applied for each miRNA passing DESeq-quality thresholds. miRNAs were considered to be differentially expressed with padj < 0.05 and not differentially expressed with padj < 0.1. However, in the EAR/no LAR group, no miRNA passed correction for multiple testing, so we applied a less stringent threshold of nom. pval < 0.001 for correction. Using Venn diagram analysis (Venny 2.1.0), significantly dysregulated (*p* < 0.001) miRNAs of the EAR plus LAR and EAR/no LAR groups were compared [76].

### 4.8. miRNA Validation by Taqman qPCR

The NGS-based expression levels of the following miRNAs were validated by TaqMan qPCR (TaqMan^®^ Advanced miRNA Assays; Thermo Fisher, Dreieich, Germany) as previously described: miR-15a-5p, miR-15b-5p, miR-32-5p, miR-96-5p, miR-146a-5p, miR-374a-5p, miR-378c, miR-1299, miR-4452, and miR-6832-3p [75,77]. miR-24-3p was used as an endogenous control due to its low variance and high abundance in the control samples. The cDNA preparation was performed by elongating 5 ng t-RNA at the 3’ end of the mature transcripts by poly(A) addition. The 5’ ends were extended by adapter ligation, followed by universal reverse transcription and amplification using the TaqMan^®^ Advanced miRNA cDNA Synthesis Kit and a thermal cycler (GeneAmp Cycler PCR Systems 9700 v3.12., Thermo Fisher, Dreieich, Germany). The thermocycler program for the RT cycles was programmed according to the manufacturer’s instructions. For qPCR, 5 µL of a 1:10 (*v*:*v*, in TE buffer)-diluted cDNA was added to the oligonucleotide. The qPCR was performed using StepOnePlus Real-Time PCR systems and StepOnePlus™ Software v2.3 (Thermo Fisher Scientific, Dreieich, Germany). The evaluation was performed according to the 2^−ΔΔCt^ method in Expression Suite Software v1.1 (Fisher Scientific, Dreieich, Germany).

### 4.9. KEGG-Pathway and Target Analysis

KEGG-pathway enrichment analysis based on microRNA signature was performed using miRNet (miRTarBase v8.0) for all identified miRNAs to identify the target genes of each miRNA [78]. Pathways associated with asthma and inflammation, TGF-β signaling, Toll-like receptor signaling, focal adhesion, and T cell receptor signaling were investigated for targets that are simultaneously regulated by three miRNAs, miR15a-5p, miR-15b-5p, and miR-374a-5p.

### 4.10. Statistics

Statistical analyses were performed using RStudio 1.2.1335 (https://cran.r-project.org/, accesses on 17 June 2021), GraphPad Prism 9.5.1 software (GraphPad software, La Jolla, CA, USA, accessed on 4 September 2023). Student’s *t*-test or the Mann–Whitney test were used depending on the normality and homogeneity of variance assumptions, respectively. Differences between multiple groups were tested using one-way ANOVA with Bonferroni corrected post-hoc analysis or corresponding non-parametric test. For correlation analysis, the Spearman or non-parametric Pearson testing was used. Statistical significance was defined as *p* < 0.05.

## 5. Conclusions

The miRNA profiles of patients with both EAR plus LAR significantly differ from patients with EAR only. The identification of several proinflammatory miRNAs suggests their involvement in chronic airway remodeling, supporting the hypothesis of a LAR-endotype associated with a higher risk for a more severe disease course.

## Figures and Tables

**Figure 1 ijms-25-01356-f001:**
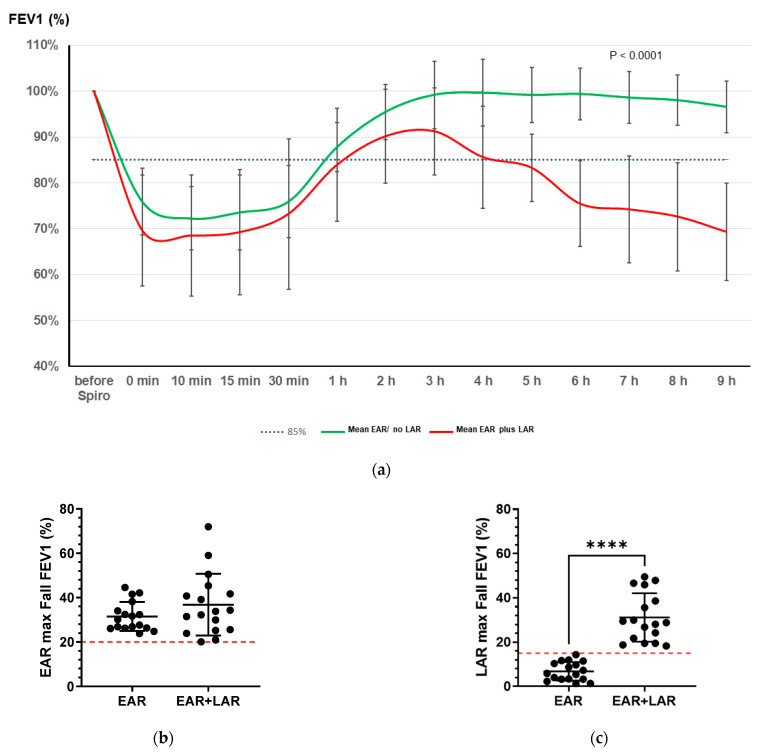
FEV1 (maximum fall) after bronchial allergen provocation (BAP). (**a**) Time course of FEV1 (%) measurements in patients with house dust mite allergy (HDM) following bronchial HDM allergen provocation (BAP), distinguishing between those exhibiting an EAR/no LAR (green line) and an EAR plus LAR (red line). (**b**) Maximum fall in FEV1 (%) during EAR, and (**c**) maximum fall in FEV1 (%) in the EAR/no LAR group (EAR) and the EAR plus LAR group (EAR + LAR). EAR was defined as a decrease of 20% in FEV1, and LAR was defined as a decrease in FEV1 ≥ 15%, occurring 6–9 h after BAP (dotted red line; black circles represent patients). **** *p* < 0.0001.

**Figure 2 ijms-25-01356-f002:**
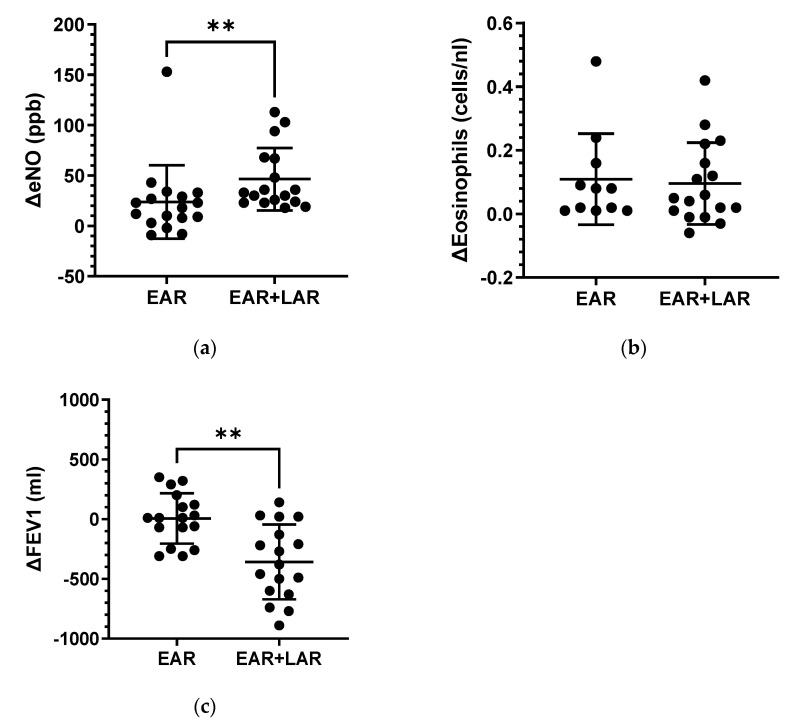
Delta changes in exhaled nitric oxide (ΔeNO), eosinophil count in peripheral blood, and delta FEV1 levels in patients with HDM following BAP. (**a**) Delta exhaled NO (eNO), (**b**) Delta eosinophils in peripheral blood, and (**c**) Delta FEV1 values are compared between patients with EAR/ no LAR (EAR) and those with both EAR and LAR (EAR + LAR), calculated from pre- and post- BAP measurements. Black circles represent patients. ** *p* < 0.01.

**Figure 3 ijms-25-01356-f003:**
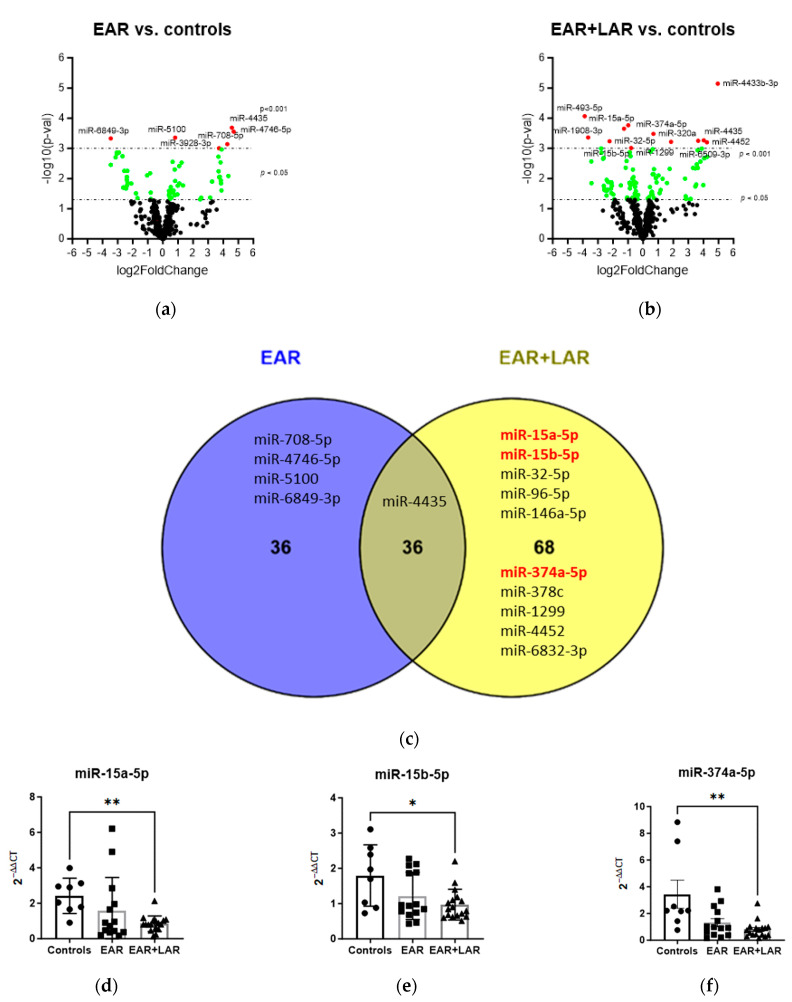
Influence of BAP on miRNA expression. Volcano plots illustrating miRNA expression profiles in patients with house dust mite allergy (HDM) 7 h after bronchial allergen provocation (BAP), depicting (**a**) those with an EAR/no LAR (EAR) and (**b**) those with EAR plus LAR (EAR + LAR) compared to a healthy control group. Significantly dysregulated miRNAs are represented by green dots (*p* < 0.05) and red dots (*p* < 0.001). (**c**) Venn diagram analysis (Venny 2.1.0) was used to compare miRNAs with a significance level of *p* < 0.05, with miRNAs at *p* < 0.001 listed in the diagram. miRNAs listed in red and bold were verified by qPCR. Expression levels of (**d**) miR15a-5p, (**e**) miR-15b-5p, and (**f**) miR-374a-5p in patients with HDM allergy displaying EAR/no LAR (EAR; black squares), EAR plus LAR (EAR + LAR; black triangles) 6–9 h after BAP, and healthy controls (black circles) as analyzed by qPCR. * *p* < 0.05, ** *p* < 0.01.

**Figure 4 ijms-25-01356-f004:**
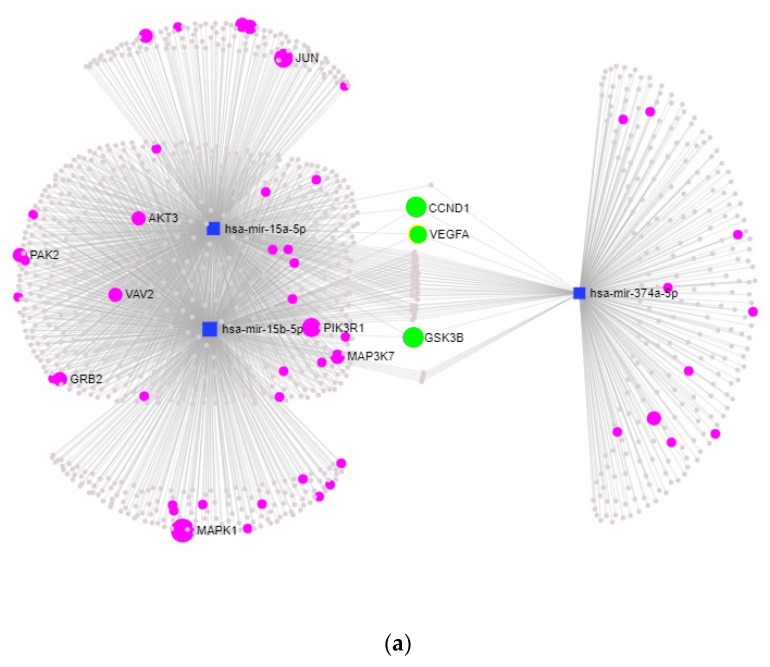
Interaction and target analysis. (**a**) Network plot illustrating Kyoto Encyclopedia of Genes and Genomes (KEGG) pathway and target analysis for miR-15a-5p, miR-15b-5p, and miR-374a-5p, performed using miRNet (miRTarBase v8.0). Targets involved in (**b**) TGF-β signaling, (**c**) focal adhesion, (**d**) Toll-like receptor signaling, and (**e**) T cell receptor signaling are displayed. In the network plots, miRNAs are represented as blue squares, targets as gray dots, targets involved in inflammatory pathways as purple dots, and targets regulated by all three miRNAs as green dots.

**Figure 5 ijms-25-01356-f005:**
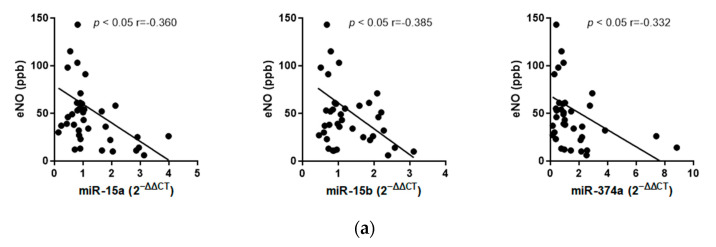
Correlation of miRNA expression with lung function and allergic inflammation. The correlations between the expression of miR-15a-5p-, miR-15b-5p-, and miR-374a-5p with (**a**) exhaled nitric oxide (eNO), (**b**) eosinophils in peripheral blood, and (**c**) the maximum fall in LAR (FEV1%) are illustrated.

**Figure 6 ijms-25-01356-f006:**
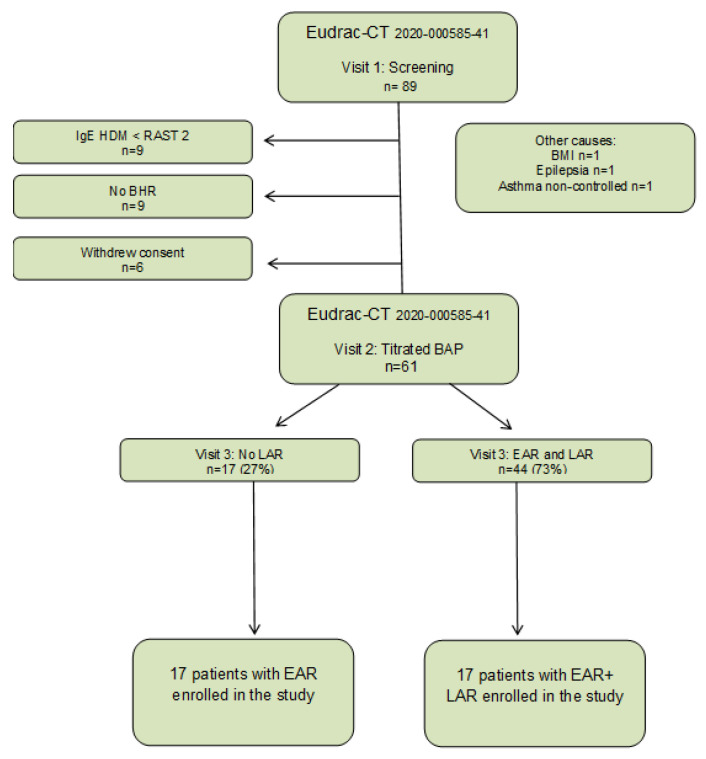
Study flow chart.

**Table 1 ijms-25-01356-t001:** Characteristics of patients.

	Controls	EAR/ No LAR	EAR Plus LAR
Numbers	*n* = 8	*n* = 17	*n* = 17
Age (years)	25 (18–35)	27 (18–38)	24 (19–32)
Sex male/female	3/5	12/5	6/11
Asthma control test (ACT)	-	23 (16–25)	23 (15–25)
eNO (ppb)	13.5 (6–36)	18 (8–74)	21 (7–104)
FEV1 (%)	98.5 (90.0–107.6)	93.4 (79.2–131.4)	97.9 (82.6–115)
Metacholine PD20 (mg)	1.3 (0.11–2.90)	0.68 (0.01–2.69)	0.48 (0.01–1.33)
Total-IgE (KU/L)	298 (57–691)	156 (28–947)	148 (4–985)
Dpt- specific IgE (KU/L)	7.5 * (0–13.0)	6.38 (0.4–21.6)	15.05 * (1.21–63.3)
Dfa-specific IgE (KU/L)	2.2 * (0.5–16.8)	7.85 (0.48–37.1)	18.15 * (0.57–65.1)

* *p* < 0.05 between controls, EAR/no LAR and EAR plus LAR. Data are shown as median and range.

## Data Availability

The data presented in this study are available on request from the corresponding author. The data are not publicly available due to privacy and ethical concerns.

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
