# Peer review of "MicroRNA Profiling of the Inflammatory Response after Early and Late Asthmatic Reaction"

_ijms, 2024, doi:10.3390/ijms25021356_

Round 1
Reviewer 1 Report
Comments and Suggestions for Authors
The current article focuses on understanding the role of microRNAs in the inflammation process associated with allergic asthma, particularly in response to HDM allergens. The article is comprehensive and offers valuable insights into the epigenetic regulation of asthma. However, there are areas where the authors might consider addressing or clarifying further:
-
-The methodology section should clearly articulate the control measures taken to ensure the reliability of results. For instance, the article could benefit from more detailed descriptions of the control groups used in the study and how they compare to the test subjects.
-
-While the article provides statistical data, a more detailed explanation of the statistical methods used for analyzing the miRNA data could enhance the readers' understanding. This would include a more thorough description of the statistical tests employed and the rationale behind their selection.
-
-The article should discuss the size and demographic representation of the study sample in more detail. First of all, sample was sample size estimated and how? Please specify it in data analysis section.
-
-While the study does a good job of discussing the biological implications of the findings, it could also elaborate on the potential clinical applications. This includes discussing how these findings could influence future asthma treatments or diagnostic methods.
-
-The article might benefit from a section discussing potential avenues for future research. This could include investigating the mechanisms by which the identified miRNAs influence asthma pathology or exploring therapeutic strategies targeting these miRNAs.
-
- I found several english errors throughout the manuscript. Please have deep language revision.
moderate revision required
Author Response
Please see the attachement.

Reviewer 2 Report
Comments and Suggestions for Authors
The paper “MicroRNA profiling of the inflammatory response after early and late asthmatic reaction” aimed to analyze miRNA profile in patients with mild asthma and HDM-allergy after bronchial allergen provocation (BAP). The article conforms to readers' interest of IJMS with sufficient data. However, there are some shortcomings that need to be further improved or explained.
Comments:
Q1. The innovation expressions of this paper should be further improved, especially in the last paragraph of introduction. Please supplement the miRNA profiling of the EAR and LAR in patients in previous papers.
Q2. The resolution of Figure 1(a) should be further improved.
Q3. In Figure 1 and 2, the data of control group is missing.
Q4. The effects of miR-15a-5p, miR-15b-5p, and miR-374a-5p on exact signaling pathway should be further identified or analyzed, if possible, please supplement the relevant proteins expressions.
Round 2
Reviewer 1 Report
Comments and Suggestions for Authors
Authors replied to my comments in a satisfactorily way. Ok to accept.
Author Response
Dear Reviewer#1,
We want to express our sincere gratitude for your thorough review of our manuscript and for acknowledging the satisfactory nature of our responses to your comments. Your constructive feedback has been invaluable in improving the quality of our work.
Thank you once again for your valuable contribution to the peer review process.
Kind regards,
Ruth Duecker